# Assessing Knowledge, Beliefs, and Behaviors around Antibiotic Usage and Antibiotic Resistance among UK Veterinary Students: A Multi-Site, Cross-Sectional Survey

**DOI:** 10.3390/antibiotics11020256

**Published:** 2022-02-16

**Authors:** Sarah E. Golding, Helen M. Higgins, Jane Ogden

**Affiliations:** 1School of Psychology, Faculty of Health and Medical Sciences, Stag Hill Campus, University of Surrey, Guildford GU2 7XH, UK; j.ogden@surrey.ac.uk; 2Institute of Infection, Veterinary and Ecological Sciences, University of Liverpool, Neston, Cheshire CH64 7TE, UK; h.higgins@liverpool.ac.uk

**Keywords:** antimicrobial resistance, veterinary students, veterinary education, beliefs, other-blaming, antimicrobial stewardship, one health

## Abstract

Antimicrobial resistance (AMR) is a profound threat to human and animal health. Antimicrobial prescribing behaviours are influenced by psychological factors such as knowledge, beliefs, and emotions. As future antimicrobial prescribers, it is important to understand beliefs about AMR and stewardship among veterinary (vet) students. This cross-sectional online survey assessed vet students’ self-reported behavior, knowledge, and beliefs in specific relation to antibiotic resistance (ABR) and antibiotic usage. Participants were early years (first- and second-year; *n* = 460) and later-years (third- and fourth-year; *n* = 113) undergraduate vet students from three UK universities. Self-reported antibiotic-related behaviors were responsible among most students. Knowledge about ABR and stewardship was moderate among early years students and good among later years students. Vet students typically believed that vets had less responsibility for both causing and preventing ABR than other groups (animal owners, human medics, and the public). This study offers evidence that vet students (along with other groups) tend to lay greater responsibility for ABR/AMR outside of their own profession, which may impact their future prescribing behaviors. It is vital that AMR and antimicrobial stewardship are embedded across veterinary curricula, and that the One Health nature of the challenge posed by AMR is emphasized to encourage shared responsibility across all stakeholder groups, thereby helping to reduce ‘other-blaming’ for AMR.

## 1. Introduction

Antimicrobial resistance (AMR) is a global, One Health problem, which poses a profound threat to human and animal health [1,2,3]. Tackling increasing rates of AMR, which are at least partly driven by inappropriate prescribing by doctors, veterinarians (vets) and other prescribers [4,5,6,7,8,9], will involve developing an understanding of what drives prescribing behaviors among these different groups.

A growing body of evidence shows that prescribers, in both human and animal medicine, are influenced by a range of psychological, social, and environmental factors [9,10]. Such factors include emotions [11,12,13], habit [14,15], the actions of colleagues [12,16], poor infection prevention and control (IPC) or biosecurity measures [17,18,19], and local and national policies [20,21,22,23]. Prescribing behaviors are also influenced by prescribers’ beliefs and perceptions about risk [24,25,26] and their beliefs about patients’ or clients’ expectations [24,27,28,29].

Despite reasonably good awareness of the need for antimicrobial stewardship among prescribers [10,24], there is evidence that prescribers only take partial responsibility for AMR and stewardship, which may, in part, be due to the beliefs that prescribers hold about other groups’ antimicrobial-related behaviors. For example, research with both farm vets and companion animal vets has shown that they psychologically distance themselves from the issue of AMR and lay the blame for AMR, and the need for increased stewardship, with other groups, especially human medics and the general public [22,28,30,31,32]. Human medics (e.g., doctors, nurse prescribers, pharmacists) also engage in this psychological distancing and other-blaming for the problem of AMR; they too believe greater stewardship is needed by other groups outside of their own profession and blame others for exerting pressure on them to prescribe [33,34,35,36].

In addition to understanding the beliefs held by practicing vets and doctors about their responsibilities for AMR and stewardship, it is important to understand the knowledge and beliefs that veterinary medicine students (vet students) and human medicine students (medical students) hold about AMR. Vet and medical students represent the future of their respective professions, and upon entering clinical practice they will need to be daily advocates of antimicrobial stewardship. Indeed, evidence suggests that more recently qualified vets have greater awareness of the threats from AMR and the need for increased stewardship than more senior vets [18,37,38]. Despite this, junior vets report feeling less confident in their client relationships and, therefore, less confident in their ability to challenge clients’ expectations for antimicrobials [38,39,40]. Junior vets and doctors also report times when they feel unable to challenge senior colleagues or prescribing norms [9,12,26,28,39,40,41,42]. This suggests that although newly qualified vets and doctors might wish to promote stewardship, they face considerable interpersonal barriers to implementing their stewardship ideals in practice.

It is therefore important to consider the beliefs and experiences of vet and medical students, and to identify ways of better supporting them to prescribe in line with stewardship principles as they transition into their practitioner roles. For both medical and vet students, the nature and content of stewardship training varies across undergraduate and postgraduate training [43,44,45]. Vet and medical students from Europe and Australia rep ort different levels of preparedness for performing responsible antimicrobial prescribing once they qualify [46,47,48]. Medical students would like more feedback on the appropriateness of their prescribing choices during training [49], and more education on drug selection and combination therapy [50]. Indeed, medical and vet students, as well as newly qualified vets and doctors, consistently report wanting more training on antimicrobial stewardship and responsible prescribing [46,48,49,51,52,53,54,55,56,57,58].

Surveys of medical students have found that knowledge, beliefs, and behaviors related to AMR and antimicrobial usage vary, with evidence of incorrect knowledge and beliefs, and less responsible behaviors, by medical students. For example, 35% of medical students from four universities in Pakistan admitted to self-medication with antimicrobials [59], while 40% of undergraduate students in China reported having self-medicated with antimicrobials in the previous six months [60]. Furthermore, although medical students had better (but still only moderate) knowledge than non-medical students about antimicrobials in the Chinese survey, medical students were more likely than non-medical students to self-medicate [60]. In France, medical students demonstrated high awareness of AMR but many students reported a lack of confidence surrounding responsible prescribing, especially regarding the selection of the correct drug and dose [53]. Only 40% of medical students in the USA were familiar with the term ‘antimicrobial stewardship’, although awareness varied between the three surveyed medical schools [54]. Across one survey of UK vet, medical, nursing, pharmacy, and dentistry students, only 44% had heard the terms ‘antimicrobial stewardship’ or ‘antibiotic stewardship’, and only a fifth felt they had sufficient knowledge about antimicrobials for their future practice [56].

Similar variation in knowledge, beliefs, and behavior around responsible antimicrobial usage has been found in surveys of vet students. For example, while vet students in Bangladesh generally had better knowledge about AMR and stewardship than their non-medical student peers, 29% of vet students reported self-medicating with antimicrobials and only 56% reported completing the full course of treatment if they started feeling better [61]. Vet students in Australia, Croatia, and Serbia demonstrated mixed knowledge about the appropriateness of different antimicrobial classes as first-line treatment and whether systemic antimicrobials were indicated across a range of clinical scenarios [46,51]. Knowledge levels about AMR and stewardship vary from good to poor among vet students in Nigeria, Sudan, and South Africa [57,58,62,63]. In Nigeria, only a minority of vet students demonstrated adequate competency across a range of clinical vignettes [57], although self-reported behavior for own use of antibiotics was reasonably responsible [58]. Whilst all vet students in a UK survey knew bacteria could become resistant to antibiotics, 28% and 34%, respectively, believed (incorrectly) that humans and animals could also become resistant to antibiotics [56]. This survey also identified that only two-thirds of vet students had heard of the British Veterinary Association’s seven-point plan for the responsible use of antimicrobials in animals [56]. 

### Aims

Existing surveys of medical and vet students, therefore, provide evidence of gaps in their knowledge and beliefs about AMR and stewardship, and less responsible behavior by students regarding use of antimicrobials for treating themselves. However, the number of surveys conducted with vet students to date is limited, with only one other published survey being conducted in the UK context [56]. Furthermore, most surveys have not explicitly explored vet students’ beliefs about different groups’ levels of responsibility for AMR and stewardship. This current study, therefore, sought to contribute to the evidence base by assessing self-reported behavior, knowledge, and beliefs about AMR and antimicrobial usage in UK undergraduate veterinary medicine students.

To avoid potential confusion or lack of understanding in terminology by students who had yet to start any pharmacology training, study materials explicitly referred to antibiotics and antibiotic resistance (ABR) instead of AMR. As such, the hypotheses, materials, and analyses refer to antibiotics and ABR. A cross-sectional online survey with first- and second-year undergraduates (early years students) from three UK universities, and with third- and fourth-year undergraduates (later years students) in one of these universities, was conducted to achieve the following aims: To describe self-reported antibiotic usage behavior among early and later years vet students.To describe knowledge levels about ABR and antibiotic use among early and later years vet students.To describe beliefs among early and later years vet students about different groups’ responsibilities for both preventing and causing ABR.To explore potential differences in vet students’ behavior, knowledge, and beliefs between early and later years students within one university.

## 2. Results

The survey link was accessed by 593 participants (vet students) from three UK universities; 16 participants did not progress beyond the information sheet and consent form, and two participants completed demographic information only. The completed responses from two other participants were excluded: one person reported being in their fifth-year (i.e., their clinical placement year) and were from another university, and one person reported being aged between 12 and 17 years old. This left a final sample size of 573 participants who completed at least the questions about antibiotic-related behavior; 534 participants completed all four measures. The knowledge scale was completed by 566 participants, the responsibility for causing ABR scale by 545 participants, and the responsibility for preventing ABR scale by 543 participants.

Participants were early years (first and second-year) undergraduate vet students (*n* = 460) from the Universities of Bristol (*n* = 237), Liverpool (*n* = 65), and Surrey (*n* = 158). Later years (third and fourth-year) students at the University of Surrey were also recruited (*n* = 113). Based on estimates for the number of students in each cohort, response rates were early years at Bristol, 79%; early years at Liverpool, 20%; early years at Surrey, 63%; and later years at Surrey, 75%. Response rates were lower at Liverpool due to a high-profile city-wide event on the day of recruitment that affected transport in the city and lecture attendance. Most participants described themselves as female (*n* = 469) and their ethnicity as white (*n* = 516), and most were aged 18–24 years old (*n* = 526). See Table 1 for full sample demographics.

### 2.1. Demographic Checks

Cell frequencies were examined for demographic variables by year group and by university. Except for year of study by university, no variable met requirements for minimum expected cell counts. Demographic baseline differences between year groups and between universities were, therefore, assessed using Fisher’s exact test (except for year of study by university, which was assessed using Pearson’s chi-square test). There were no statistically significant differences observed between year groups or universities at baseline on any demographic variable (details in Appendix A).

### 2.2. Behavior

#### 2.2.1. Early Years

Among early years vet students at Bristol, Liverpool, and Surrey, most reported responsible use of antibiotics for the three questions asked (Table 2). The vast majority reported never taking antibiotics to help them get better more quickly from a cold (86.5%) or to prevent symptoms from getting worse when they had a cold (90.0%). Although most early years students reported never stopping taking antibiotics when they started feeling better (78.7%), 15.0% did report that they sometimes, often, or always stopped taking antibiotics when they started feeling better. The mean behavioral scores were low for both symptom management and treatment cessation (Table 3), indicating most early years students reported never performing these behaviors. There were no statistically significant differences observed in self-reported behavior among early years students between universities for either symptom management, *H*(2) = 5.26, *p* = 0.072, *S* = 3.78, or treatment cessation, *H*(2) = 1.82, *p* = 0.40, *S* = 1.32.

#### 2.2.2. Later Years

Among later years vet students at Surrey, almost all reported responsible use of antibiotics for all three questions (Table 2). No later years students responded often or always to the symptom management questions, 92.9% reported never taking antibiotics to help them get better more quickly from a cold and 94.7% reported never taking antibiotics to prevent symptoms from getting worse when they had a cold. Most later years students reported never stopping taking antibiotics when they started feeling better (85.8%), with only 7.1% reporting that they sometimes, often, or always stopped taking antibiotics when they started feeling better. The mean behavioral scores for both behavioral measures were low (Table 3). Most later years students at Surrey reported never performing these behaviors. 

#### 2.2.3. Differences in Behavior between Early and Later Years

Despite lower mean scores for both behavioral measures among Surrey later years students compared to Surrey early years students, there was no statistically significant difference observed in self-reported symptom management behavior (*U* = 8622.50, *z* = 0.84, *p* = 0.40, *S* = 1.32, *r* = 0.05) or self-reported treatment cessation behavior (*U* = 8463.00, *z* = 1.12, *p* = 0.26, *S* = 1.94, *r* = 0.07) between the two groups of students (*n* = 271).

### 2.3. Knowledge

#### 2.3.1. Early Years

Knowledge about antibiotics and ABR was categorized in this study as moderate among early years students. Mean knowledge across early years vet students was 6.67 (maximum score = 8; Table 4), with 58.7% providing the correct answer for seven or eight of the eight knowledge items (good knowledge) and 37.7% providing the correct answer for five or six items (moderate knowledge). The number of correct items ranged from three to eight (Figure 1). Almost all early years students knew ABR could threaten human and animal welfare (99.6%) and that misuse of antibiotics could lead to ABR (99.8%). Knowledge scores were lower regarding the spread of ABR and potential harm from antibiotic treatment: 33.1% of early years students did not know that ABR can spread between bacteria and 39.7% did not know that antibiotic treatment can be harmful to patients (Table 5). There were no statistically significant differences observed in knowledge between early years students from different universities, (*H*(2) = 4.12, *p* = 0.13, *S* = 2.94).

#### 2.3.2. Later Years

Knowledge about antibiotics and ABR among later years students was categorized as good. Mean knowledge was 7.25 (Table 4), with 82.3% providing the correct answer for seven or eight knowledge items, and 16.9% providing the correct answer for five or six items; the number of correct items ranged from four to eight (Figure 1). There were three apparent knowledge gaps among later years students: 15.0% of later years students did not correctly identify that ‘antibiotic resistance’ does not describe humans becoming immune to antibiotics, 20.3% did not know that antibiotics are not useful for colds and flu, and 23.0% did not know that antibiotic treatment may harm patients (Table 5). 

#### 2.3.3. Differences in Knowledge between Early and Later Years

Knowledge differed between early and later years students at Surrey (*n* = 267). Later years students were observed to have better knowledge than early years students (*U* = 10,936.50, *z* = 3.77, *p* < 0.001, *S* = 9.97, *r* = 0.23)

### 2.4. Beliefs about Responsibility for Causing ABR

#### 2.4.1. Early Years

There were no statistically significant differences in beliefs between universities on three subscales, but there were statistically significant differences in beliefs for the Public/Patients subscale (details in Appendix A). 

To explore whether early years students believed that any group might be more or less responsible for causing ABR, a Friedman’s ANOVA was conducted. Across early years students, there were statistically significant differences observed in vet students’ beliefs about the level of responsibility between groups for causing ABR, *χ*^2^(3) = 277.60, *p* < 0.001, *S* = 9.97. These differences were explored using pairwise comparisons with adjusted *p* values. 

Early years students appeared to believe that vets were less responsible for causing ABR compared to animal owners, human medics, and the public/patients (all *p* < 0.001, *S* = 9.97). Differences in mean beliefs about these different groups represented medium effect sizes (Table 6 and Table 7). Early years students also believed the public/patients were more responsible for causing AMR compared to human medics (*p* = 0.045, *S* = 4.47), although this was represented by a very small effect size. They did not appear to believe there was any difference in responsibility for causing ABR between animal owners and human medics (*p* = 1.00, *S* = 0.00), or between animal owners and the public/patients (*p* = 0.42, *S* = 1.25). Early years vet students believed that the public/patients have most responsibility for causing ABR, followed by animal owners and human medics, with vets having the least responsibility for causing ABR (Figure 2).

#### 2.4.2. Later Years

Mean levels of later years students’ beliefs about which groups have responsibility for causing ABR are reported in Table 6. Across later years students, there were statistically significant differences observed in beliefs about the level of responsibility between groups for causing ABR, *χ^2^*(3) = 91.35, *p* < 0.001, *S* = 9.97. 

Later years students believed vets were less responsible for causing ABR compared to animal owners, human medics, and the public/patients (all *p* < 0.001, *S* = 9.97). Differences in beliefs about these different groups ranged from medium to large effect sizes (Table 7). Later years students also believed the public/patients were more responsible for causing ABR than human medics (*p* = 0.001, *S* = 9.97), which was represented by a small effect size. Later years students did not appear to believe there was any difference in responsibility for causing ABR between animal owners and human medics (*p* = 0.11, *S* = 3.18) or between animal owners and the public/patients (*p* = 0.99, *S* = 0.01). Later years vet students believed the public/patients have most responsibility for causing ABR, followed by animal owners and human medics, with vets having the least responsibility for causing ABR (Figure 2). 

#### 2.4.3. Differences in Beliefs between Early and Later Years

There were no statistically significant differences observed between early and later years students at Surrey regarding their beliefs about the amount of responsibility each of the four groups have for causing ABR (details in Appendix A).

### 2.5. Beliefs about Responsibility for Preventing ABR

#### 2.5.1. Early Years

There were no statistically significant differences observed in beliefs between universities on two subscales, but there were statistically significant differences in beliefs for the Public/Patients subscale and the Animal Owners subscale (details in Appendix A). 

Across early years students there were statistically significant differences observed in vet students’ beliefs about the level of responsibility between groups for preventing ABR *χ*^2^(3) = 219.04, *p* < 0.001, *S* = 9.97. Early years students believed vets were less responsible for preventing ABR compared to animal owners, human medics, and the public/patients (all *p* < 0.001, *S* = 9.97). Differences in mean beliefs about these different groups represented small to medium effect sizes (Table 6 and Table 7). Early years students also believed animal owners were more responsible for preventing ABR compared to human medics (*p* < 0.001, *S* = 9.97) and the public/patients (*p* < 0.001, *S* = 9.97); these differences were both small effect sizes. They did not appear to believe there was any difference in responsibility for preventing ABR between human medics and the public/patients (*p* = 1.00, *S* = 0.00). Early years vet students believed animal owners have most responsibility for preventing ABR, followed by the public/patients, then human medics, with vets having the least responsibility for preventing ABR (Figure 3).

#### 2.5.2. Later Years

Mean levels of later years students’ beliefs about which groups have responsibility for preventing ABR are reported in Table 6. Across later years students, there were statistically significant differences in beliefs about the level of responsibility between groups for preventing ABR *χ*^2^(3) = 19.49, *p* < 0.001, *S* = 9.97. Later years students believed vets were less responsible for preventing ABR than animal owners (*p* = 0.001, *S* = 9.97); this difference was a small effect. All other pairwise comparisons did not demonstrate statistically significant differences (Table 7), suggesting later years students believed groups mostly have similar levels of responsibility for preventing ABR, but that animal owners have more responsibility for preventing ABR than do vets (Figure 3). 

#### 2.5.3. Differences in Beliefs between Early and Later Years

Beliefs about the amount of responsibility three of the four groups have for preventing ABR did not appear to statistically significantly differ between early and later years vet students (details in Appendix A). There was, however, a statistically significant difference in beliefs about the level of responsibility that students believed animal owners have for preventing ABR (*U* = 6283.50, *z* = 2.67, *p* = 0.008, *S* = 6.97, *r* = 0.17). Early years students had higher mean beliefs (mean = 4.52) that animal owners were responsible for preventing ABR than did later years students (mean = 4.45). 

## 3. Discussion

This cross-sectional survey with UK undergraduate vet students assessed their self-reported behavior, knowledge, and beliefs about ABR and antibiotic usage. Key findings were that behavior was considered mostly responsible among both early and later years students, early years students had moderate knowledge while later years students had good knowledge, and on average, both early and later years students believed vets had less responsibility for causing and preventing ABR than all other groups. 

Self-reported antibiotic usage behaviors were judged to be responsible in most early years students across all three universities (Bristol, Liverpool, Surrey), and in most later years students at Surrey, with no statistically significant differences observed in self-reported behavior between the early and later years students at Surrey. There were, however, some students who reported inappropriate usage behaviors. Among all early years students, 21.3% reported stopping antibiotic courses when they started to feel better, and 13.5% reported taking antibiotics to help them feel better more quickly when they had a cold. Among later year students, these proportions were 14.2% and 7.1%, respectively. Other surveys have also found that vet and medical students report sometimes engaging in less responsible behaviors, such as self-medicating with antimicrobials, ceasing treatment early, or sharing antimicrobials with other people [56,58,59,60,61].

Knowledge about ABR was better among the later years students at Surrey than among the early years students at Surrey. Three key knowledge gaps identified among both early and later years vet students were not knowing that antibiotics are not useful for colds and flu, not knowing that human and animal patients can be harmed from antibiotic treatment, and not knowing that ABR does not describe humans becoming immune to antibiotics. Two further knowledge gaps among early years students were not knowing that ABR refers to how bacteria avoid being killed by antibiotics, and not knowing that ABR can spread between bacteria. There is some evidence that, as would be expected, medical and veterinary curricula do have the desired effect of improving knowledge about responsible use of antimicrobials [58,60,61]. Nonetheless, the consistent picture across surveys of vet and medical students and newly qualified vets and doctors is that there remain gaps in knowledge and confidence around AMR and stewardship [46,48,49,51,52,53,54,55,56,57,58,64]. This body of evidence suggests further work is required to strengthen the AMR and antimicrobial stewardship elements of vet and medical training. 

On average, both early years and later years vet students believe that vets have less responsibility for causing ABR than do animal owners, human medics, and the public. Early years students also typically believe that vets have less responsibility for preventing ABR than these other three groups. Later years students typically believe that vets have the same responsibility for preventing ABR as do human medics and the public, but that animal owners have more responsibility for preventing ABR. Similar patterns of beliefs about responsibility for AMR were found in a previous survey of practicing farm vets that posed similar questions to those used in this current study [32]. As with practicing doctors and vets, who often lay the blame for AMR and poor stewardship behaviors with other groups [22,28,30,31,32,33,34,35,36], vet students in this current study also appear to consider their own (future) prescribing to be less of a contributor to AMR than that of other prescribers. Similar findings have been found in other surveys of vet and medical students [56,57,65]. Together, these findings suggest that both practicing prescribers and students appear to locate more of the blame for ABR/AMR, and greater responsibility for stewardship, with other groups. This other-blaming for the issue of AMR is not, however, specific to the medical and veterinary professions; there is also evidence that other stakeholder groups, such as the public (as patients), farmers, and pet owners all engage in some level of other-blaming for inappropriate prescribing and usage of antimicrobials [19,28,36,66,67]. 

### 3.1. Limitations

Some limitations to this study relate to the finding that knowledge was better among later years students. First, as measures were taken cross-sectionally, rather than longitudinally as students progressed through their course, it is possible differences in knowledge between early and later years students may be a cohort effect rather than a potential increase in knowledge as students have progressed with their studies. Second, later years students were only recruited at Surrey, and not at Bristol or Liverpool; had data been collected for later years students across all three universities, greater confidence could have been placed in the findings, assuming the same difference in knowledge was observed between the early and later years students across all three universities. Third, no assessment was made regarding curriculum content at any of the three universities. Through informal discussions with veterinary medicine lecturers at the universities, it was understood that early years students would not have covered AMR or responsible antimicrobial usage in much detail. As curriculum content was not assessed, however, it is possible that the differences in knowledge between the early and later years students at Surrey may have been due to differences in curriculum content. 

Additional limitations relate to the generalizability of the findings to other vet students. There are currently eight universities in the UK running undergraduate veterinary medicine training, so the findings from this survey may not generalize across the other five vet schools. Furthermore, these findings are likely to be less generalizable outside of a UK context, as training courses for veterinarians in other countries can have different lengths and curriculum content compared to UK courses.

Finally, there are limitations relating to the measures themselves, so findings from this study should be considered as tentative and exploratory, with further work needed to help support or refute these findings. First, as with all studies using self-report measures, there is the risk that participants’ responses were influenced by social desirability bias; for example, vet students’ actual antibiotic-related behavior may not be as responsible as they reported. Second, the measures used to assess behavior, knowledge, and beliefs in this study have not been subject to psychometric validation (e.g., face validity with a range of experts, factor analysis, test-retest reliability), so there is a risk that these measures do not, in fact, assess the constructs they are intended to assess. As in-depth face validity was not conducted, there is a risk that items used may be ambiguously worded or confusing to participants, and borderline values for Cronbach’s alpha also indicate potentially inadequate internal consistency. Additionally, the 12-item scales used to assess beliefs in this study were reduced from the 24-item scales used in a previous study through considered discussion rather than through statistical analysis. Nonetheless, the use of non-validated or partially validated scales to measure psychological and behavioral constructs related to AMR appears to be relatively common, with examples of fully validated scales rare [68,69,70]. Future research should, therefore, look to conduct psychometric validation of existing measures, to strengthen the measurement of psychological and behavioral constructs of relevance to AMR, and to determine whether such constructs may be unidimensional or multidimensional. 

### 3.2. Implications

Given the One Health nature of AMR [2,3,71], and the tendency towards other-blaming by both vets and human medics, potential implications from this study are discussed with reference to both veterinary and human medical training. Although the specific context of prescribing might be different for vets and doctors (as it would also be between specialisms within vet or human medicine), evidence increasingly suggests that prescribers from all domains of medicine are influenced by psychological and social factors such as emotions, prescribing norms, and economics [9,10]. An interdisciplinary, One Health approach to understanding the drivers of antimicrobial stewardship could help researchers, prescribers, educators, and other stakeholders recognize that, despite contextual differences, there are also shared similarities in the factors that influence prescribing and stewardship behaviors. By recognizing these similarities, insights from different domains of medicine can potentially be used to inform and guide developments across all domains of medicine [3]. 

The findings from this cross-sectional survey of vet students at three UK universities provide evidence among vet students for other-blaming for the problem of ABR. This pattern of other-blaming for ABR/AMR and less responsible stewardship behaviors has previously been identified in other studies with vet and medical students [56,57,63,65], and with practicing vets and doctors [22,28,30,31,32,33,34,35,36]. Educators need to be aware of the tendency among both students and practicing prescribers to locate the problem of ABR /AMR and the responsibility for stewardship with other groups, especially as there is tentative evidence that these beliefs may be related to prescribing behaviors [32]. One possible intervention to help reduce some of this tendency to other-blame could be to embed a One Health approach across curricula for vet and medical students, especially in areas where there is considerable potential for overlap between veterinary and human medicine. One Health is especially relevant for AMR, but also for other areas of health including mental health, zoonotic diseases, environmental degradation, and injury prevention [71]. 

Vet and medical students will inevitably develop their professional identities as vets or as doctors during their training and early careers, but if a One Health approach is embedded across healthcare curricula this might help reduce the salience of specific professional identities as students transition to prescribers and they begin to encounter clinical situations that they learnt about within a One Health framework. There is tentative evidence that this would be welcomed by students, especially vet students. In a UK survey, 69% of vet students, and 42% of pharmacy students reported wanting more education on the links between human, animal, and environmental health [56], and in an Australian survey, vet students had positive attitudes towards One Health approaches in both training and practice [72]. 

By framing AMR and stewardship as a shared, One Health issue—or *common fate* [73]—across all domains of medicine, and by increasing collaborative working between different human, animal, and environmental health sectors, the tendency for other-blaming could be reduced. Social identity theory [74,75] posits that people are motivated to act to protect the reputation of groups with which they identify, which could partially explain why vet students (and other groups) tend to lay greater responsibility for AMR with other groups. By increasing the salience of a common fate across groups (the shared threat from AMR), members of these groups may start to identify as one larger group, increasing co-operation to achieve a common goal of improved stewardship and thereby reducing other-blaming for AMR [73,75]. Social identity theory is increasingly being applied to health-related behaviors and outcomes [76], and future research could explore the potential value of social identity approaches to help drive changes in prescribing and stewardship behaviors. 

Finally, this study indicates that there are some knowledge gaps about ABR among UK vet students that educators should ensure are addressed. Other studies highlight that vet and medical students feel there are gaps in their education relating to responsible antimicrobial prescribing and stewardship [46,49,50,51,52,53,54,55,56,57,58] and not all medical or vet students feel suitably prepared to practice responsible prescribing [46,47,50]. Educators could, therefore, look to increase the focus on AMR and antimicrobial stewardship to improve vet and medical students’ knowledge as they progress through their training, and to increase their confidence in performing best practice stewardship behaviors once they qualify. For example, there is evidence that introducing a dedicated curriculum on antimicrobial stewardship can have positive influences on knowledge and attitudes about AMR among medical and pharmacy students, as well as improving self-efficacy beliefs towards engaging in inter-professional collaboration [77]. AMR knowledge and confidence in antimicrobial prescribing appear to be higher when medical students have undertaken an infectious disease rotation or have had frequent contact with infectious disease specialists [54,65]. There is also evidence from an intervention with medical students in the Netherlands that an e-learning course designed to improve antimicrobial prescribing choices increased knowledge scores and drug choice in a later simulated clinical examination [78]. Universities should ensure that antimicrobial stewardship and an understanding of AMR is embedded in all teaching related to the selection and prescribing of antimicrobials in undergraduate courses for vet and medical students, as current course content on antimicrobial stewardship can vary [43,44]. Indeed, making antimicrobial stewardship a core component of all training, at all levels including continuous professional development, is a key recommendation in the WHO’s global action plan on AMR [1].

## 4. Materials and Methods

### 4.1. Design

Veterinary students (vet students) at three English universities completed a cross-sectional survey (hosted online in Qualtrics) to assess their self-reported behaviors, knowledge, and beliefs in relation to antibiotic use and ABR. 

### 4.2. Participants

Participants were early years (first and second-year) undergraduate vet students from the Universities of Bristol, Liverpool, and Surrey, and later years (third and fourth-year) students from the University of Surrey. These three universities were selected for reasons of access to potential participants based on existing relationships with colleagues across these institutions; the research team were based across Surrey and Liverpool and had links to colleagues at Bristol, meaning the team could opportunistically recruit from students at these three universities. Furthermore, these universities represented a geographical spread across England and included both long-established vet schools (Liverpool and Bristol) and the UK’s newest vet school at the time (Surrey). 

During planning, the intention was to only sample early years students across Bristol, Liverpool, and Surrey. At a late stage of planning, the opportunity arose to sample later years students at Surrey, so the decision was made to also recruit from this cohort. Due to the timelines at this point in the project, it was decided not to risk any agreed permissions regarding approval to recruit, or potentially delay agreed recruitment dates, at Bristol and Liverpool, as the goal was to ensure that all recruitment took place during the same semester across all three universities. The decision was therefore made to not seek additional permissions to recruit from the later years students at Bristol and Liverpool. 

There were no restrictions to participation on the basis of any demographic variable except age; participants needed to be aged 18 or over. The only other inclusion criterion was that participants were currently in years one to four of their undergraduate course (as students in their fifth year would be undertaking clinical placements and would be expected to have more detailed knowledge regarding ABR, prescribing, and stewardship).

Recruitment took place between October and December 2018. Participants were recruited by lecturers (Liverpool and Bristol) or by SG (Surrey) during lectures. Students were verbally invited to take part in an online survey exploring their beliefs about medicines use (it was not revealed during recruitment that the survey was specific to antibiotics). It was stressed that participation was voluntary and not linked to course performance or credits. Students were provided with a short url (displayed on lecture hall screens using PowerPoint) to enter into their own mobile or laptop devices; they were instructed to read the participant information sheet and complete the consent form if they were willing to complete the survey. Students were advised to ask the recruiter if they had any questions. 

#### Power Calculation

As antibiotic-related knowledge, beliefs, and behaviors had not been previously explored across different groups of vet students at the time the study was designed, estimating an effect size for potential difference in these variables between groups was problematic. Therefore, the power calculations, instead, explored whether there was a reasonable chance of recruiting sufficient participants to detect a range of different effect sizes. 

Prior to recruitment, it was estimated that across the early years cohorts there were approximately 300 students at Bristol, 330 students at Liverpool, and 250 students at Surrey. It was expected around 80% of students would attend any given lecture (based on HH’s experience of teaching undergraduate vet students). As the recruiter would be present in the room to advertise the opportunity to participate and explain the potential benefits of participating (i.e., supporting their research training and professional development by gaining experience of being a research participant), and that time to complete the survey (if they wished to) was being provided during scheduled lecture time, it was assumed that around 80% of those in attendance (i.e., most students) would be likely to complete the survey (estimated as: Bristol *n* = 192, Liverpool *n* = 211, Surrey *n* = 160). Therefore, from a total population of approximately 880 students, it was expected that a minimum sample of 563 participants would be recruited across the early years cohorts. The later years cohort at Surrey was smaller (as this is a recently established vet school) with an estimated 150 students. Following the same logic, it was estimated that 96 students might be recruited from this cohort. 

Power calculations conducted using G*Power [79] indicated the following sample sizes for detecting different effects using one-way ANOVAs: detecting a large effect (*η*^2^ = 0.25) would require a minimum sample of 33 participants; detecting a medium effect (*η*^2^ = 0.09) would require a minimum of 102 participants, and detecting a small effect (*η*^2^ = 0.01) would require a minimum of 957 participants.

### 4.3. Measures

All study materials were presented online using Qualtrics, and were accessible using desktops, laptops, and mobile devices (see Appendix B for survey questions). The survey had been piloted with eight psychology postgraduate researchers to establish it could reasonably be completed in under 10 minutes (who were also asked to provide feedback if any items were unclear; none did). This was to ensure that recruitment and survey completion had only a minimal impact on students’ contact time with their lecturers.

#### 4.3.1. Demographics

Participants were asked about their age, gender, and ethnicity (they could decline to provide this information), as well as their university and year of study for their veterinary degree. Age was requested in age bands to help preserve anonymity. 

#### 4.3.2. Antibiotic Behavior

Self-reported behavior was assessed using a three-item scale developed for this study (see Appendix B, Table A1). Participants were asked to indicate how often they tended to perform three types of antibiotic-related behaviors on a scale of 1 (*never*) to 5 (*always*). The three behavioral statements used as items in this scale were identified from a systematic review and meta-analysis that explored and synthesized statements about self-reported antibiotic-related behaviors commonly put to the general public in surveys about antibiotic use and knowledge [80]. 

Cronbach’s alpha (*α*) for the three behavioral items was 0.49; the inter-item correlation matrix showed two items correlated highly together (Q1, “When I get a cold, I will take antibiotics to help me get better more quickly” and Q2, “When I get a cold, I will take antibiotics to prevent my symptoms from getting worse”; *r* = 0.69), but the third item (Q3, “I normally stop taking antibiotics when I start feeling better”) was poorly correlated with these two items (Q1 *r* = 0.23; Q2 *r* = 0.21). When Q3 was removed, Cronbach’s *α* = 0.82. Therefore, two behavioral outcome measures were created for the analysis: symptom management behavior (Q1 and Q2), and treatment cessation behavior (Q3). For this study, behavior was defined as: responsible = answering *never*, inappropriate = answering *occasionally*, *sometimes*, *often*, or *always*.

#### 4.3.3. Knowledge about Antibiotic Use and ABR

Knowledge was assessed using an eight-item scale developed for this study (see Appendix B, Table A2) based upon previous surveys about antibiotic knowledge. The general public have been extensively surveyed about their knowledge and beliefs around antibiotic use and ABR [81,82,83] as have, to a lesser extent, medical and veterinary students [46,49,54,84]. In these surveys, whilst medical and veterinary students are often asked similar questions to those put to the public, they are additionally asked questions specific to their clinical training. For this survey, first- and second-year vet students would have had little or no exposure to microbiology and clinical decision-making training, so it was deemed more appropriate to assess their knowledge in a similar way to assessing knowledge among the general public. Therefore, two systematic reviews of surveys of the general public [80,85] were used to develop this knowledge scale, with additional statements added for this study. 

The scale assessed four areas of knowledge: (1) what should antibiotics be used for, (2) what is ABR, (3) what contributes to ABR, and (4) what are the harms from antibiotic use and ABR? Participants were asked to respond to eight statements about antibiotic use and resistance, with three possible responses: *true*, *false*, or *don’t know*. Example items are: “Antibiotics are useful for colds and flu” and “Antibiotic resistance can spread between bacteria”. Knowledge scores were created by totaling the number of correct answers provided by participants (maximum score = eight). For this study, knowledge level was defined as: good = seven or eight items correct, moderate = five or six items correct, and poor = four or less items correct. 

#### 4.3.4. Beliefs about Responsibility for Causing and Preventing ABR

Beliefs about different groups’ responsibility for causing ABR and preventing ABR were measured using two 12-item scales (see Appendix B, Table A3 and Table A4) that were adapted from two 24-item scales previously developed for measuring these beliefs in an earlier study with practicing farm vets [32]. Each 12-item scale presented to vet student participants in this current study comprised four subscales, with three items about each of four target groups: vets, animal owners, human medics, and human patients. For both scales, participants were asked “to what extent do you think each of the following contributes to causing/preventing antibiotic resistance? If you are unsure, please give your best guess”. Participants responded on a scale of 1 (*contributes not at all*) to 5 (*contributes very much*). 

The 12-item responsibility for causing ABR scale measured the extent to which participants thought the behavior of different groups (including themselves, as future vets) contributes to increasing rates of ABR. Statements focused on antibiotic use by each group. Example items participants were asked to rate are: “The number of antibiotic prescriptions that GPs write”, “Patients not completing their antibiotic courses”, and “Animal owners using antibiotics to treat viral infections in their animals”. Internal consistency for three of the four causing ABR subscales was below the threshold for acceptability for the early stages of research (0.6) [86] but could not be improved by removing any of the items (Table 8).

The 12-item responsibility for preventing ABR scale measured the extent to which participants thought all groups are equally responsible for preventing ABR. Statements focused on what members of each group could do differently to help reduce selective pressure for ABR. Example items participants were asked to rate are: “Members of the public taking antibiotics as instructed by their doctors”, “Vets using more diagnostic tests”, and “Animal owners accepting that their animals don’t always need antibiotics”. Internal consistency for all four preventing ABR subscales was acceptable (Table 1).

The two original 24-item scales that were administered to practicing farm vets in a previous study [32] were developed based on interviews conducted with farm vets [30]. These 24-item scales, which assess beliefs about different groups’ responsibilities for causing and preventing AMR, present a series of statements about antibiotic use by six groups (human medics, human patients, companion animal vets, pet owners, farm animal vets, and farmers). These groups were identified by farm animal vets as being key stakeholder groups in promoting antimicrobial stewardship. Item generation for the original scales was also informed by reviewing relevant literature to capture different types of behavior that can potentially drive AMR. In this current cross-sectional study with vet students, the existing 24-item scales were adapted to merge farmers and pet owners into one group, and farm vets and companion animal vets into another group (resulting in items about four, rather than six, stakeholder groups). Additionally, in the original two 24-item scales used previously [32], four items were presented for each of the six groups in question; in the two 12-item versions used in the current study, three items were presented for each of the four groups in question. These adaptations were made for three reasons: (1) as vet students are not yet specialized as either farm or companion animal vets, drawing a distinction between these groups was less important than in the study with practicing farm vets, (2) to adapt the language to focus on antibiotics, rather than antimicrobials, as vet students may not yet be familiar with the use of the term ‘antimicrobials’ in this context, and (3) to minimize the length of the survey for practical reasons, as survey recruitment and completion took place during lectures. 

### 4.4. Procedure

Participants first saw the participant information sheet and consent form, followed by a very brief instructions page. They then provided demographic details before being presented with questions about: (1) self-reported behaviors regarding antibiotics; (2) knowledge about antibiotics and ABR, and (3) beliefs about which groups are responsible for causing and preventing ABR. Finally, participants were thanked for taking part and presented with a debrief statement. The whole study took an average of about six minutes for participants to complete. 

#### Randomization

To control for order effects, the order of presentation for all items within each scale was randomized. Additionally, the order of presentation of the responsibility for causing ABR and responsibility for preventing ABR scales was also randomized, to control for any potential contamination between these scales. All randomization was performed using Qualtrics. 

### 4.5. Data Analysis

Missing data were not imputed; cases with missing data were excluded pairwise in all analyses. ‘Forced entry’ was used for all items in the main measures in Qualtrics, meaning missing data only occurred when participants declined to continue the survey; there were no missing data for individual items on completed scales. Analysis was conducted using IBM SPSS Statistics version 24. 

Checks for group differences between years and universities were performed using Pearson’s chi-square test and Fisher’s exact test for the categorical demographic variables. Kruskal-Wallis tests were used to check for group differences in behavior, knowledge, and beliefs between universities for the early years students, as all scores were not normally distributed. Statistically significant results (*p* < 0.05) were followed up using pairwise comparisons. Friedman’s ANOVAs were used to compare within-participant differences in beliefs about different groups for both the early years sample and the later years sample. These tests were also followed up using pairwise comparisons with adjusted *p* values. Potential differences between early and later years students’ behavior, knowledge, and beliefs were explored using Mann-Whitney tests within the Surrey cohort data only (these scores were also not normally distributed).

Alongside *p*-values, 95% confidence intervals, and effect sizes, the *S*-value (Shannon Information or binary surprisal) is also reported, as it is considered a more intuitive metric for interpreting evidence against the test hypothesis; in this study, the null hypotheses [87,88,89]. The *S*-value is the base-2 logarithmic transformation of the *p*-value, which transforms the *p*-value into a measure of information (in bits; binary digits) against a test hypothesis; the greater the *S*-value, the less compatible the observed data are with the test hypothesis [87,88,89]. By rounding the S-value to the nearest integer, it transforms probabilities associated with test statistics into a measure of information that can be understood in terms of the amount of information that would be gained from the same number of coin tosses [87,88,89]. For example, an *S*-value of 2 indicates the observed test statistic is about as surprising as tossing two heads in two fair coin tosses, whereas an *S*-value of 10 would indicate the test statistic is about as surprising as tossing 10 heads in 10 fair coin tosses. 

### 4.6. Ethics

Informed consent was taken and recorded online via Qualtrics. Participants needed to indicate agreement with all required statements before commencing the survey. Consent records and survey data are held electronically in line with university requirements and current data protection regulations. 

Participants were advised that taking part was voluntary, that they could choose not to participate without providing a reason and were given the researchers’ contact details. Due to the nature of data collection (a large group completed the survey simultaneously, and each survey response was anonymous, with no identifiable personal data collected), there was no way of offering the chance to withdraw data once participants had commenced the survey (although they could choose not to continue to the end). This was made clear in the information sheet and consent form. 

The study was self-assessed by SG for ethical approval on 10 September 2018 (reference: 353003-352994-39062693; amendments completed on 4 October 2018, reference: 353003-352994-39791323), in line with the University of Surrey’s ethical review procedures. Appropriate approval was also gained from the Universities of Bristol (reference: Study 3017) and Liverpool (reference: VREC687) to recruit their veterinary students. The study has been reported in line with the STROBE checklist [90].

## 5. Conclusions

This cross-sectional online survey of UK-based vet students assessed their self-reported behavior, knowledge, and beliefs in relation to ABR and antibiotic usage. Findings suggest that although self-reported behavior was responsible among vet students there were some gaps in their knowledge about ABR. Furthermore, results also indicated that, on average, vet students believe vets have less responsibility for causing and preventing ABR than other groups. Vet students typically appear to locate more of the responsibility for ABR and stewardship with animal owners, human medics, and the public. This pattern of other-blaming has been found in other research with medical students and practicing vets and doctors. Efforts should be directed towards embedding ABR/AMR and antimicrobial stewardship training more deeply across curricula for veterinary students. Adopting a One Health approach to training and continuous professional development that emphasizes shared responsibility for ABR/AMR and antimicrobial stewardship across human, animal, and environmental health, may help reduce other-blaming between groups and increase co-operation towards a shared goal of improving stewardship.

## Figures and Tables

**Figure 1 antibiotics-11-00256-f001:**
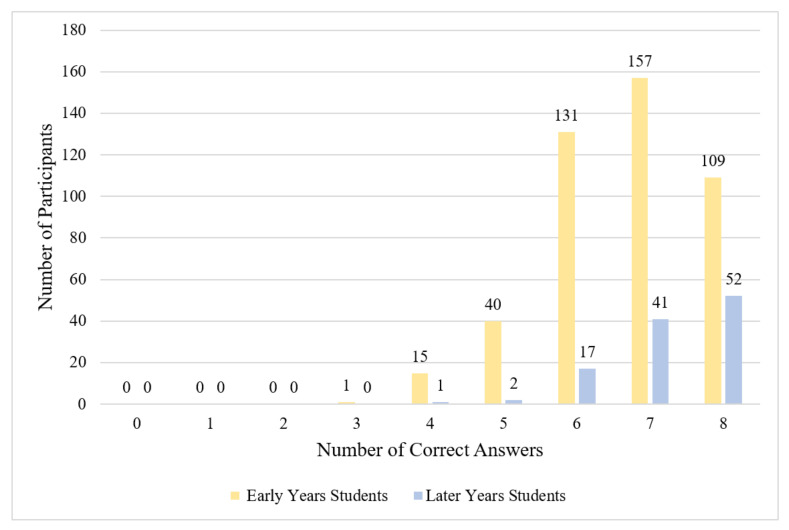
Frequency of knowledge scores (early years, *n* = 453; later years *n* = 113).

**Figure 2 antibiotics-11-00256-f002:**
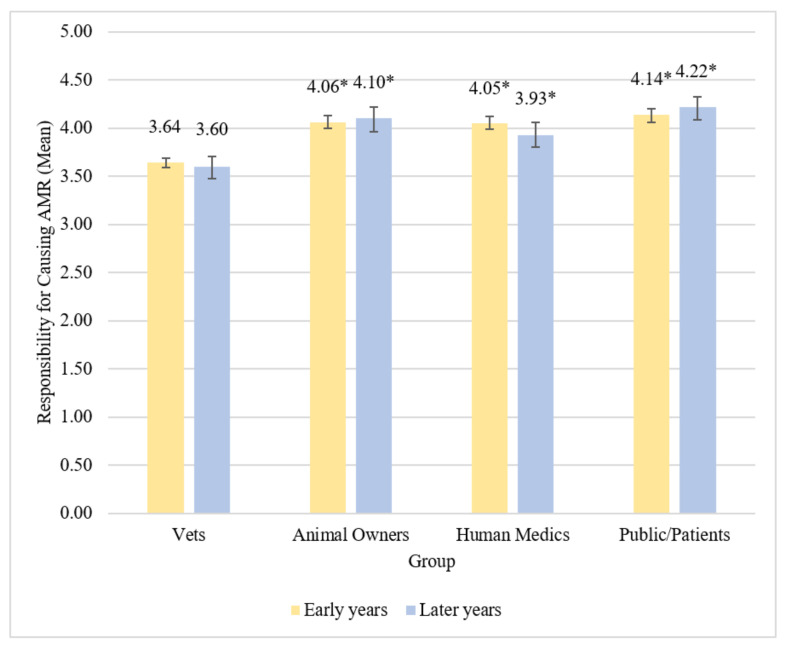
Mean beliefs about responsibility for causing ABR by group among early and later years students. Vets designated as reference group. * denotes difference from reference group at *p* < 0.001. Error bars represent 95% bias-corrected and accelerated bootstrap confidence intervals.

**Figure 3 antibiotics-11-00256-f003:**
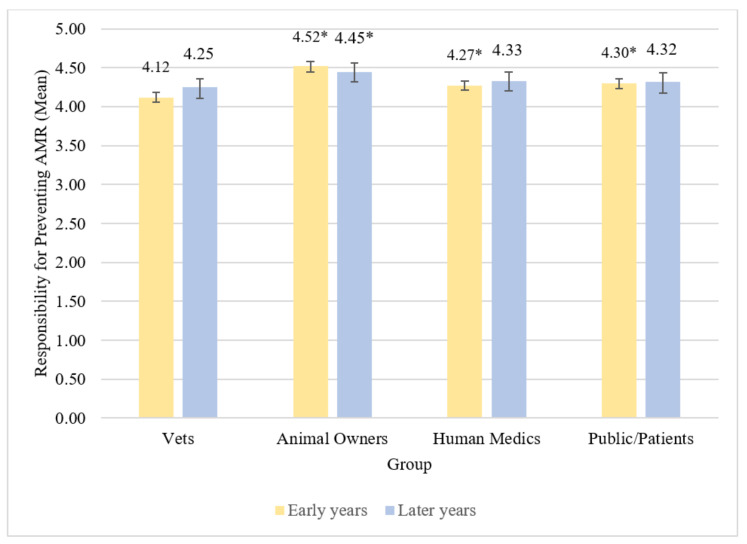
Mean beliefs about responsibility for preventing ABR by group among early and later years students. Vets designated as reference group. * denotes difference from reference group at *p* < 0.001. Error bars represent 95% bias-corrected and accelerated bootstrap confidence intervals.

**Table 1 antibiotics-11-00256-t001:** Demographic details for whole student sample.

Characteristic	Individuals (*n* = 573)	Percentage of Sample
Age (Years)		
18–24	526	91.80
25–34	37	6.46
35–44	4	0.70
45–54	1	0.17
55–64	2	0.35
65 and Over	1	0.17
Prefer Not to Say	2	0.35
Gender		
Female	469	81.85
Male	99	17.28
Other	1	0.17
Prefer Not to Say	4	0.70
Ethnicity ^1^		
White	516	90.05
Black	1	0.17
Asian	23	4.01
Mixed	26	4.54
Other	3	0.52
Prefer Not to Say	4	0.70
University ^1^		
Bristol	237	41.36
Liverpool	65	11.34
Surrey	271	47.29
Year of Study		
First	260	45.38
Second	200	34.90
Third (Surrey Only)	60	10.47
Fourth (Surrey Only)	53	9.25

^1^ Percentages do not exactly total 100% due to rounding.

**Table 2 antibiotics-11-00256-t002:** Frequency of self-reported behaviors.

Item	Number of Participants (Percentage of Sample)									
	Early Years Students (*n* = 460)					Later Years Students (*n* = 113)				
	Never	Occasionally	Some-Times	Often	Always	Never	Occasionally	Sometimes	Often	Always
Symptom Management										
When I get a cold, I will take antibiotics to help me get better more quickly ^1^	398 (86.52)	41 (8.91)	15 (3.26)	6 (1.30)	0 (0.00)	105 (92.92)	7 (6.19)	1 (0.88)	0 (0.00)	0 (0.00)
When I get a cold, I will take antibiotics to prevent my symptoms from getting worse	414 (90.00)	29 (6.30)	9 (1.96)	7 (1.52)	1 (0.22)	107 (94.69)	5 (4.42)	1 (0.88)	0 (0.00)	0 (0.00)
Treatment Cessation										
I normally stop taking antibiotics when I start feeling better	362 (78.70)	29 (6.30)	27 (5.87)	12 (2.61)	30 (6.52)	97 (85.84)	8 (7.08)	2 (1.77)	2 (1.77)	4 (3.54)

^1^ Percentages do not exactly total 100% due to rounding.

**Table 3 antibiotics-11-00256-t003:** Mean behavior scores.

			Early Years by University							
	Early Years Whole Sample (*n* = 460)		Bristol (*n* = 237)		Liverpool (*n* = 65)		Surrey (*n* = 158)		Later Years Surrey (*n* = 113)	
Measure	Mean (SD)	BCa 95% CI	Mean (SD)	BCa 95% CI	Mean (SD)	BCa 95% CI	Mean (SD)	BCa 95% CI	Mean (SD)	BCa 95% CI
Symptom Management	1.18 (0.50)	1.13, 1.22	1.14 (0.42)	1.10, 1.20	1.28 (0.61)	1.16, 1.44	1.18 (0.56)	1.10, 1.27	1.07 (0.25)	1.03, 1.12
Treatment Cessation	1.52 (1.14)	1.42, 1.63	1.54 (1.18)	1.40, 1.70	1.62 (1.20)	1.36, 1.90	1.45 (1.06)	1.30, 1.62	1.30 (0.89)	1.17, 1.45

Note. Bootstrap results are based on 1000 bootstrap samples. BCa 95% CI = 95% bias-corrected and accelerated bootstrap confidence intervals. SD = Standard deviation.

**Table 4 antibiotics-11-00256-t004:** Mean knowledge scores.

			Early Years by University							
	Early Years Whole Sample (*n* = 453)		Bristol (*n* = 237)		Liverpool (*n* = 62)		Surrey (*n* = 154)		Later Years Surrey (*n* = 113)	
Measure	Mean (SD)	BCa 95% CI	Mean (SD)	BCa 95% CI	Mean (SD)	BCa 95% CI	Mean (SD)	BCa 95% CI	Mean (SD)	BCa 95% CI
Knowledge	6.67 (1.05)	6.57, 6.77	6.58 (1.01)	6.43, 6.72	6.81 (1.02)	6.57, 7.05	6.74 (1.12)	6.57, 6.91	7.25 (0.84)	7.08, 7.40

Note. Bootstrap results are based on 1000 bootstrap samples. BCa 95% CI = 95% bias-corrected and accelerated bootstrap confidence intervals. SD = Standard deviation.

**Table 5 antibiotics-11-00256-t005:** Rates of correct knowledge per item.

	Number of Participants (Percentage of Sample)			
	Early Years Students (*n* = 453)		Later Years Students (*n* = 113)	
Item (Correct Answer)	Correct	Incorrect or Don’t Know	Correct	Incorrect or Don’t Know
Antibiotics can treat bacterial infections (true)	432 (95.36)	21 (4.64)	111 (98.23)	2 (1.77)
Antibiotics are useful for colds and flu (false)	366 (80.79)	87 (19.21)	90 (79.65)	23 (20.35)
‘Antibiotic resistance’ describes how bacteria avoid being killed by antibiotics (true)	377 (83.22)	76 (16.78)	104 (92.04)	9 (7.96)
‘Antibiotic resistance’ describes humans becoming immune to antibiotics (false)	366 (80.79)	87 (19.21)	96 (84.96)	17 (15.04)
Misuse of antibiotics can lead to antibiotic resistance (true)	452 (99.78)	1 (0.22)	113 (100.00)	0 (0.00)
Antibiotic resistance can spread between bacteria (true)	303 (66.89)	150 (33.11)	106 (93.81)	7 (6.19)
Patients (both humans and animals) may be harmed from antibiotic treatment (true)	273 (60.26)	180 (39.74)	87 (76.99)	26 (23.01)
Antibiotic resistance could threaten both human and animal welfare (true)	451 (99.56)	2 (0.44)	112 (99.12)	1 (0.88)

**Table 6 antibiotics-11-00256-t006:** Mean beliefs about groups’ responsibility for causing and preventing ABR.

			Early Years by University							
	Early Years Whole Sample		Bristol		Liverpool		Surrey		Later Years Surrey	
Measure	Mean (SD)	BCa 95% CI	Mean (SD)	BCa 95% CI	Mean (SD)	BCa 95% CI	Mean (SD)	BCa 95% CI	Mean (SD)	BCa 95% CI
Responsibility for Causing ABR ^1^										
Human Medics	4.05 (0.67)	3.99, 4.12	4.02 (0.65)	3.94, 4.10	4.08 (0.77)	3.85, 4.30	4.09 (0.66)	3.99, 4.20	3.93 (0.73)	3.80, 4.06
Public/Patients	4.14 (0.69)	4.06, 4.20	4.08 (0.66)	3.99, 4.17	4.00 (0.85)	3.80, 4.19	4.29 (0.65)	4.17, 4.40	4.22 (0.64)	4.09, 4.33
Vets	3.64 (0.66)	3.59, 3.69	3.61 (0.60)	3.54, 3.68	3.64 (0.78)	3.45, 3.84	3.69 (0.69)	3.56, 3.80	3.60 (0.63)	3.48, 3.71
Animal Owners	4.06 (0.69)	4.00, 4.13	4.04 (0.65)	3.95, 4.13	3.91 (0.84)	3.68, 4.13	4.16 (0.69)	4.05, 4.29	4.10 (0.71)	3.96, 4.22
Responsibility for Preventing ABR ^2^										
Human Medics	4.27 (0.68)	4.21, 4.33	4.20 (0.73)	4.10, 4.31	4.34 (0.71)	4.15, 4.50	4.35 (0.59)	4.25, 4.44	4.33 (0.67)	4.20, 4.45
Public/Patients	4.30 (0.67)	4.23, 4.36	4.23 (0.71)	4.13, 4.32	4.29 (0.75)	4.09, 4.49	4.42 (0.55)	4.32, 4.51	4.32 (0.66)	4.17, 4.44
Vets	4.12 (0.68)	4.06, 4.18	4.09 (0.70)	4.00, 4.17	4.16 (0.73)	3.98, 4.34	4.17 (0.64)	4.07, 4.26	4.25 (0.67)	4.11, 4.36
Animal Owners	4.52 (0.63)	4.45, 4.58	4.47 (0.66)	4.37, 4.56	4.42 (0.75)	4.20, 4.60	4.64 (0.52)	4.56, 4,72	4.45 (0.63)	4.32, 4.56

Note. Bootstrap results are based on 1000 bootstrap samples. BCa 95% CI = 95% bias-corrected and accelerated bootstrap confidence intervals. df = Degrees of freedom. SD = Standard deviation. ^1^ Sample sizes: early years sample *n* = 436, Bristol *n* = 233, Liverpool *n* = 59, Surrey *n* = 144, later years Surrey *n* = 106. ^2^ Sample sizes: early years sample *n* = 435, Bristol *n* = 234, Liverpool *n* = 58, Surrey *n* = 143, later years Surrey *n* = 106.

**Table 7 antibiotics-11-00256-t007:** Pairwise comparisons for beliefs about groups’ responsibility for causing and preventing ABR.

	Early Years Students					Later Years Students				
Comparison	Mean Diff.	Std. Test Statistic	*Adj. p*	*S*	*r*	Mean Diff.	Std. Test Statistic	*Adj. p*	*S*	*r*
Responsibility for Causing ABR ^1^										
Vets v. Animal Owners	−0.42	12.01	<0.001	9.97	0.41	−0.50	6.79	<0.001	9.97	0.47
Vets v. Human Medics	−0.41	11.15	<0.001	9.97	0.38	−0.33	4.43	<0.001	9.97	0.30
Vets v. Public/Patients	−0.50	13.82	<0.001	9.97	0.47	−0.62	8.18	<0.001	9.97	0.56
Animal Owners v. Human Medics	0.01	0.87	1.00	0.00	0.03	0.17	2.36	0.11	3.18	0.16
Animal Owners v. Public/Patients	−0.08	1.81	0.42	1.25	0.06	−0.12	1.39	0.99	0.01	0.10
Human Medics v. Public/Patients	−0.09	2.68	0.045	4.47	0.09	−0.29	3.75	0.001	9.97	0.26
Responsibility for Preventing ABR ^2^										
Vets v. Animal Owners	−0.40	12.74	<0.001	9.97	0.43	−0.20	3.74	0.001	9.97	0.26
Vets v. Human Medics	−0.15	4.49	<0.001	9.97	0.15	−0.08	1.71	0.52	0.94	0.12
Vets v. Public/Patients	−0.18	5.78	<0.001	9.97	0.20	−0.07	1.29	1.00	0.00	0.09
Animal Owners v. Human Medics	0.25	8.25	<0.001	9.97	0.28	0.12	2.03	0.26	1.94	0.14
Animal Owners v. Public/Patients	0.22	6.96	<0.001	9.97	0.24	0.13	2.45	0.086	3.54	0.17
Human Medics v. Public/Patients	−0.03	1.29	1.00	0.00	0.04	0.01	0.42	0.26	1.94	0.03

Note. Adj. = Adjusted. Diff. = Difference. Std. = Standardised. V. = Versus. ^1^ Sample sizes: early years *n* = 436, later years *n* = 106. ^2^ Sample sizes: early years *n* = 435, later years *n* = 106.

**Table 8 antibiotics-11-00256-t008:** Reliability statistics for beliefs measures.

Measure	Cronbach’s α
Responsibility for Causing ABR	
Human Medics	0.63
Public/Patients	0.50
Vets	0.52
Animal Owners	0.57
Responsibility for Preventing ABR	
Human Medics	0.73
Public/Patients	0.63
Vets	0.68
Animal Owners	0.75

## Data Availability

Data are available in a publicly accessible repository. The data presented in this study are openly available in UK Data Service at: http://doi.org/10.5255/UKDA-SN-855402 (Data Collection title: Cross-sectional survey about antibiotics with UK undergraduate veterinary students, 2018). Demographic data have been removed from the publicly accessible file to ensure participant anonymity.

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
