# Peer review of "Assessing Knowledge, Beliefs, and Behaviors around Antibiotic Usage and Antibiotic Resistance among UK Veterinary Students: A Multi-Site, Cross-Sectional Survey"

_antibiotics, 2022, doi:10.3390/antibiotics11020256_

Round 1
Reviewer 1 Report
Comments
Line 3: Authors should replace “Amongst” with “Among” and in other places within the manuscript.
This manuscript focused entirely on Antibiotics and Antibiotic Resistance. I advised that these terms feature in the title rather than “Antimicrobial Usage” and Antimicrobial Resistance”
Lines 57 & 68: Please delete “students” after “vet”
Line 70: Please replace “transition” with a verb, may be “transit”.
Lines 74 – 76: Do the authors mean vet and medical students (especially the undergraduates) are licensed to prescribe antimicrobials? Please clarify. Authors should make it clearer that these students are not yet prescribing drugs.
Line 112: Please delete “and Hypotheses” from the sub-heading.
Lines 127 & 411: Please check that the font characteristics conform to those of other parts of the manuscript.
Results
The authors are fond of stating why a particular non-parametric (Mann-Whitney/Kruskal-Wallis) test was used for an analysis throughout the result section. Authors should delete these parts as they appear boring and are not meant for this section. Once a comprehensive report of the analysis carried out in this study is stated under the data analysis sub-heading, there is no need for the repetition.
Also, re-stating the use of the Freidman’s ANOVA while attempting to state the results for significance testing among later years students is unnecessary in the results section.
Authors should please replace “meaningful” with “significant” as a more appropriately word for presenting the result of difference noticed.
Please delete the “see” before all tables and figures cited within the results.
Materials and methods
Line 538: Why have the authors decided to survey only three of the 8 vet schools in the UK and only Surrey for the later years vet students? The reasons should be included in the manuscript.
More so, the authors should state in clear terms the inclusion and exclusion criteria (with reasons) used in selecting the participants across the various years studied.
Is there a possible reference to the logic used in estimating the sample size used in this survey?
Lines 716-718: Authors should indicate the approval numbers from the Institutional Review Boards in the manuscript.
Author Response
Thank you for taking the time to review our manuscript, and for your helpful feedback. We have addressed your specific comments below.
Line 3: Authors should replace “Amongst” with “Among” and in other places within the manuscript.
- We have amended accordingly in the title and throughout the manuscript (except for the reference list).
This manuscript focused entirely on Antibiotics and Antibiotic Resistance. I advised that these terms feature in the title rather than “Antimicrobial Usage” and Antimicrobial Resistance”
- Thank you for comment regarding our use of these terms in the title – we have edited the title to be about “antibiotics” rather than “antimicrobials”, as we agree with you that this would more accurately reflect the focus of the survey itself. We have also edited the abstract and conclusion to reflect the AMR/ABR distinction that we have outlined in the main text (i.e. AMR in the introduction and discussion / ABR in relation to the study materials and results).
Lines 57 & 68: Please delete “students” after “vet”
- We have deleted accordingly.
Line 70: Please replace “transition” with a verb, may be “transit”.
- We would prefer to leave this sentence as currently constructed; we understand that transition can be used as a verb.
Lines 74 – 76: Do the authors mean vet and medical students (especially the undergraduates) are licensed to prescribe antimicrobials? Please clarify. Authors should make it clearer that these students are not yet prescribing drugs.
- We have amended this sentence to make it clearer that students are not yet prescribing by adding “once they qualify” to the end of the sentence. We have also added “during training” to the sentence that follows the one you have mentioned, again to make it clearer. This section now reads:
- Vet and medical students from Europe and Australia rep ort different levels of preparedness for performing responsible antimicrobial prescribing once they qualify [46–48]. Medical students would like more feedback on the appropriateness of their prescribing choices during training [49] and more education on drug selection and combination therapy [50].
Line 112: Please delete “and Hypotheses” from the sub-heading.
- We have deleted as suggested.
Lines 127 & 411: Please check that the font characteristics conform to those of other parts of the manuscript.
- Our apologies, but we cannot see where the font characteristics are differing on these two lines (or nearby). We will, of course, liaise with the copy editors about this if the manuscript is ultimately accepted for publication.
Results
The authors are fond of stating why a particular non-parametric (Mann-Whitney/Kruskal-Wallis) test was used for an analysis throughout the result section. Authors should delete these parts as they appear boring and are not meant for this section. Once a comprehensive report of the analysis carried out in this study is stated under the data analysis sub-heading, there is no need for the repetition.
- Thank you for highlighting that this information is not required within the results section, as it is repetitive. We have now edited the results throughout to remove reference to the specific test carried out each time. We have also added the brief explanation for why non-parametric tests were conducted to section 4.5 Data Analysis in the methods.
Also, re-stating the use of the Freidman’s ANOVA while attempting to state the results for significance testing among later years students is unnecessary in the results section.
- Thank you for highlighting this. We have edited out the information about the Friedman’s ANOVA in the section about the later years students.
Authors should please replace “meaningful” with “significant” as a more appropriately word for presenting the result of difference noticed.
- We have replaced “meaningful” with “statistically significant” throughout the results as suggested.
Please delete the “see” before all tables and figures cited within the results.
- Thank you for this suggestion – we have amended accordingly.
Materials and methods
Line 538: Why have the authors decided to survey only three of the 8 vet schools in the UK and only Surrey for the later years vet students? The reasons should be included in the manuscript.
- We had already briefly commented on why we had recruited from only 3 vet schools in the manuscript under 4.2 Participants, but we have added to this section to make it clearer that this was essentially an opportunistic sample as we had existing links across these three vet schools. This section now reads (with the added text in green):
- These three universities were selected for reasons of access to potential participants based on existing relationships with colleagues across these institutions; the research team were based across Surrey and Liverpool and had links to colleagues at Bristol, meaning the team could opportunistically recruit from students at these three universities. Furthermore, these universities represented a geographical spread across England and included both long-established vet schools (Liverpool and Bristol) and the UK’s newest vet school at the time (Surrey).
- We had originally planned to only survey early years (years 1 and 2) students, and all of the initial design and planning conversations (including liaising with Bristol and Liverpool vet schools regarding ethics, permission to recruit, and potential recruitment dates during Semester 1) took place on this basis. A couple of weeks before we were ready to launch, we were presented with the opportunity to also survey the later years students at Surrey, which we decided to do. Ideally, we would have of course reached out to Bristol and Liverpool to also recruit from their later years cohorts – but with time running short, we did not wish to complicate/jeopardise the recruitment of students as already planned and agreed. We therefore decided to only recruit later years students at Surrey, which we agree is a limitation of the study (and is mentioned in the limitations section). We have added a footnote to page 18 (from the section 4.2 Participants) to outline this decision.
More so, the authors should state in clear terms the inclusion and exclusion criteria (with reasons) used in selecting the participants across the various years studied.
- Apologies for this oversight. Some of this information was presented in the results, which we have now moved to section 4.2 Participants in the methods. We have also added a statement about which years were included in the survey and why to the same section.
Is there a possible reference to the logic used in estimating the sample size used in this survey?
- Unfortunately we are not able to provide a reference for this – but our logic for the attendance estimate was based on one of the author’s (HMH) 15+ years of experience in teaching undergraduate veterinary students that most of those students attend most of the time. We also assumed that as the recruiter would be in the room to announce the opportunity to participate in the study, and that time was being provided during the lecture contact time to complete the survey, that most students probably would complete the survey. We have added some text to the Power Calculation section (4.2.1) to comment on this.
Lines 716-718: Authors should indicate the approval numbers from the Institutional Review Boards in the manuscript.
- Thank you for highlighting this oversight; the references have been added to section 4.6 Ethics.

Reviewer 2 Report
Authors reported a cross-sectional survey assessed veterinarian students’ self-reported behavior and knowledge in relation to AMR. Participants were early years (first- and second-year) and later-years (third- and fourth-year) undergraduate vet students from three UK universities. Authors found that vet students typically believed that vets had less responsibility for both causing and preventing AMR than other groups (animal owners, human medics, and the public).
The work is very interesting, but too long and difficult to read and follow due to redundant information. I suggest authors to synthesize content to help readers focus results.
Title
“Assessing Knowledge, Beliefs, and Behaviors Around Antimicrobial Usage and Antimicrobial Resistance Amongst UK Veterinary Students: A Multi-Site, Cross-Sectional Survey”
Abstract
Authors must be insert some information about material and method
Introduction
Lane 34: “other prescribers” who are you referring to?
- Aims and Hypotheses
Lanes 131 and 134: It’s clear the authors intent and that “ABR” was used for students, but in the text I would continue to write “AMR”
Results
Please, substitute “ABR” with “AMR” in the text
Discussion
Discussion well done, but too long and redundant in some points. Some information has already been given in the introduction.
Author Response
Authors reported a cross-sectional survey assessed veterinarian students’ self-reported behavior and knowledge in relation to AMR. Participants were early years (first- and second-year) and later-years (third- and fourth-year) undergraduate vet students from three UK universities. Authors found that vet students typically believed that vets had less responsibility for both causing and preventing AMR than other groups (animal owners, human medics, and the public).
The work is very interesting, but too long and difficult to read and follow due to redundant information. I suggest authors to synthesize content to help readers focus results.
- Thank you for this feedback, and for taking the time to review our manuscript - we are pleased to hear you find the work interesting. We have edited sections of the results to remove some of the repetition. We have also addressed your specific comments below.
Title
“Assessing Knowledge, Beliefs, and Behaviors Around Antimicrobial Usage and Antimicrobial Resistance Amongst UK Veterinary Students: A Multi-Site, Cross-Sectional Survey”
Abstract
Authors must be insert some information about material and method
- We had already stated in the abstract that this was a cross-sectional survey, but we have added the word “online” to make clear the method of survey. Unfortunately, due to word count for the abstract we are unable to add additional details about the specific scales that were used to measure behavior, knowledge and beliefs.
Introduction
Lane 34: “other prescribers” who are you referring to?
- We have used this phrase as a ‘catch all’ to cover other groups of practitioners who have prescribing rights, such as dentists, pharmacists, and nurse prescribers. In the interests of concision, we had not listed these examples in this sentence, but we have now included two additional citations here to further support this statement (Laxminarayan et al 2013; Pinder et al 2015).
- Aims and Hypotheses
Lanes 131 and 134: It’s clear the authors intent and that “ABR” was used for students, but in the text I would continue to write “AMR”
- Thank you for this suggestion – we have reviewed this and made a few edits to our use of “ABR” versus “AMR” in the manuscript (mostly in the title, abstract, and conclusion) but we have generally retained our current use of these terms. We discussed this issue of terminology in depth when we designed the study materials and again when we were drafting the manuscript, as we were aware that using different terms throughout the manuscript could be a potential issue. We decided that we should use the broader term “AMR” when discussing the context within which the study sits (i.e., resistance to antivirals, anthelmintics, etc, as well as to antibiotics), and it is this broader threat of AMR that we wish to acknowledge with our language in the manuscript. Furthermore, some of the studies referenced throughout the introduction and discussion are not specifically focused on antibiotics/antibacterials, so it is appropriate to use the broader term AMR in these sections of the manuscript. We also decided, however, that for clarity of study design reporting, we would use “ABR” and “antibiotics” when we were referring to study materials, which as you note was the language used in the materials that were presented to participants. We have endeavoured not to use the terms “ABR” and “AMR” interchangeably in the manuscript, and we hope that it remains clear the distinction in our usage of these terms.
Results
Please, substitute “ABR” with “AMR” in the text
- As per our previous comment, we have chosen to retain our existing use of “ABR” in the methods and results.
Discussion
Discussion well done, but too long and redundant in some points. Some information has already been given in the introduction.
- Thank you – we are pleased to hear you consider the discussion to be good. We have reviewed the discussion and introduction but have not made any substantial edits here. Although some of the same studies have been cited in both sections, we feel we have mostly presented different elements of those studies in the discussion, and discussed them more specifically in relation to our findings. Where we have summarised key points from the introduction, it is because we feel they are relevant to our discussion of potential implications. If there are specific sections of text that you feel should be removed from the discussion, please do advise, and we will be happy to review again.
